# *Anisakis* spp. Larvae in Deboned, in-Oil Fillets Made of Anchovies (*Engraulis encrasicolus*) and Sardines (*Sardina pilchardus*) Sold in EU Retailers

**DOI:** 10.3390/ani10101807

**Published:** 2020-10-05

**Authors:** Giorgio Smaldone, Rosa Luisa Ambrosio, Raffaele Marrone, Marina Ceruso, Aniello Anastasio

**Affiliations:** 1Department of Agricultural Sciences, University of Naples, Federico II, via Università 100, 80055 Portici (NA), Italy; giorgio.smaldone@unina.it; 2Centro di Riferimento Regionale per la Sicurezza Sanitaria del Pescato CRiSSaP, Campania Region, Centro Direzionale is. C3–80143 Napoli (NA), Italy; anastasi@unina.it; 3Department of Veterinary Medicine and Animal Production, Unit of Food Hygiene, University of Naples Federico II, via F. Delpino 1, 80137 Napoli (NA), Italy; rosaluisa.ambrosio@unina.it (R.L.A.); marina.ceruso@unina.it (M.C.)

**Keywords:** semipreserved fish products, *Anisakis* spp., food safety, anchovies, sardines

## Abstract

**Simple Summary:**

*Sardina pilchardus* and *Engraulis encrasicolus* are largely consumed in Europe as fresh or ripened dishes. Their consumption may represent a public health risk in regard to *Anisakis* allergic reactions and anisakiasis. This study aimed to evaluate the presence of *Anisakis* spp. larvae in deboned, in-oil anchovy and sardine fillets marketed in the EU. Ninety semipreserved anchovy and sardine products were examined to evaluate the presence and viability of larvae and identify them. Only 30 nonviable anisakid larvae were found, indicating that processing technologies can influence their presence in final products. It is, however, important that visual inspection is performed only by trained people and that the sources of raw materials are considered in the production flow chart.

**Abstract:**

*Sardina pilchardus* and *Engraulis encrasicolus* are considered the principal target species for commercial fishing in Europe and are widely consumed as semipreserved products. Although they are considered shelf-stable products, if treatment is not correctly applied, their consumption may represent a public health risk in regard to anisakiasis and allergic reactions. Little is known about the prevalence of *Anisakis* spp. in ripened products. This study aimed to evaluate the presence of *Anisakis* spp. larvae in deboned, in-oil anchovy and sardine fillets marketed in the EU to assess the influence of processing techniques on the prevalence of larvae. Ninety semipreserved anchovy and sardine products deriving from the Mediterranean Sea or Atlantic Ocean were collected from different EU retailers and examined using chloropeptic digestion to evaluate the presence of larvae and identify them. Thirty nonviable Anisakid larvae—*A. pegreffii* (30%) and *A. simplex* (70%)—were found. The frequency of larvae was higher in anchovies (28.8%). The low frequency of parasites found proved that processing technologies can influence the presence of larvae in final products, but it is important that visual inspection is performed only by trained people. The sources of raw materials should be considered in the production flow chart.

## 1. Introduction

European pilchard (Sardine) *Sardina pilchardus* (Walbaum, 1792) and European anchovy *Engraulis encrasicolus* (Linnaeus, 1758) are small, valuable, pelagic species and are considered one of the principal target species for commercial fishing in Europe. Although *S. pilchardus* is more abundant, fishing pressure on the European anchovy is greater due to its higher commercial value as raw fish [1]. However, given that they are used in a wide range of traditional and homemade dishes and are frequently consumed raw, e.g., marinated in lemon juice or vinegar, pickled, or salted–ripened [2,3,4], both species are economically important. According to The European Market Observatory for Fisheries and Aquaculture Products (EUMOFA), EU landings of anchovy decreased by 9% in 2016 and the average price for anchovies increased by 9%, moving from 1.55 to 1.68 Euro/kg, while in 2016, EU landings of sardine showed an increase of 17% over 2015, with the average sardine price at the European level declining by 13% compared to 2015, dropping from 0.94 to 0.82 Euro/kg [1]. At the same time, an increase in demand and consumption of preserved or semipreserved fishery products was also registered; in particular, Europe’s average annual consumption of canned sardines increased by 30% in recent years from 0.53 to 0.69 kg/capita [1], and consumption of processed anchovies rose over the past few years (+2–3% per year) [5]. These kinds of products are considered to be intrinsically healthy and are convenient and tasty foods appreciated for their textures and characteristic odors and flavors [6,7]. Salted–ripened products are considered shelf-stable because of their low water activity levels, low moisture, high water-phase salt, and high salt contents. However, their consumption could represent a public health risk of anisakiasis, a zoonosis carried by Anisakid nematodes, if they do not undergo a process to ensure the killing of viable parasites. When humans ingest raw or undercooked fish or cephalopods harboring viable L3 larvae in the edible portion, these nematodes can cause disease and even a severe acute or chronic gastrointestinal infection and/or an IgE-mediated allergic reaction in severe cases [8,9,10]. According to The European Food Safety Authority (EFSA), *Anisakis* is one of the major parasitological risks to humans deriving from the consumption of fish [11]. In The European Union (EU) the Rapid Alert System for Food and Feed (RASFF) reported 39 notifications of the presence of parasites in 2018, 92.30% of which were accountable to *Anisakis* spp. [12]. European legislation enforces the unconditional application of preventive freezing treatment at −20 °C for at least 24 h or −35 °C for 15 h for all products consumed raw or almost raw as semipreserved fish products [13]. Because freezing cam negatively influence the organoleptic characteristics of salted products [14,15], this legislation is not always be applied, particularly in domestic contexts. Moreover, although deep freezing and other treatments such as salting ensure no viable larvae in the fish products [2,16], the possible presence of thermostable allergens in the edible part of the fish could be a risk for certain hypersensitive individuals should be highlighted [8,17,18], and visible parasites, even if nonviable, could represent a defect [19,20], thereby altering the quality of the product and making it unfit for human consumption [21,22]. The prevalence of *Anisakis* spp. in fresh anchovies and sardines from EU regions is widely reported in the literature [14,23,24,25,26,27,28], but little is known about ripened products. Guardone et al. [6], in a recent trial, focused their research on different kinds of products made from anchovies and found a high percentage of parasite larvae in whole salted–ripened products. This study aimed to assess the presence and viability of *Anisakis* spp. larvae in previously salted, deboned, in-oil fillets made of anchovies and sardines sold in the EU. Our attention was focused only on this kind of product to evaluate the influence of the processing techniques on the presence of larvae and viability.

## 2. Materials and Methods 

### 2.1. Sampling

The study was carried out on ninety previously salted, deboned, in-oil fillets. In particular, 45 samples from 25 different brands of anchovies (AN) and 45 samples from 19 different brands of sardines (SA) were collected from different EU retail outlets (product information derived from labels reported in Table 1) between March 2018 and September 2019.

To carry out proportionate convenience sampling, 23 AN and 26 SA examples from the Mediterranean Sea (ANM and SAM, respectively) were sampled, while 22 AN and 19 SA samples from the Atlantic Ocean (ANA and SAA respectively) were sampled. All samples were examined in the Food Chemistry laboratory of the Department of Veterinary Medicine and Animal Production, University of Naples Federico II. 

### 2.2. Parasitological Analysis

Each sample was assigned a progressive number, excess oil was eliminated, and chloropeptic digestion was performed on the edible part. The whole sample was digested in an ACM-11806 Magnetic Stirrer Multiplate in a chloropeptic solution as suggested by [29]. Digestions were performed for 15 minutes at an incubation temperature of 37 °C in an acid solution (pH = 1.5) with 0.063 M HCl. The assay used liquid pepsin at a concentration of 0.5% and a ratio of 1:10 sample weight/solution volume. Artificial digestion was used because it is cheap and fast with these kinds of processed products, with no differences in accuracy or specificity compared with the UV-press method [30]. The digested solution was decanted through a strainer and the digestion remains and larvae were collected and transferred into a pepsin digestion solution (0.5% w/v pepsin in 0.063 M HCl) and inspected for viability under a stereomicroscope at 37 °C for one hour [20]. All nematode larvae were placed in individual Eppendorf tubes with 70% ethanol for further molecular analysis.

### 2.3. Anisakis Larvae Identification

All Anisakid larvae were identified by microscopic examination of diagnostic characters [31] at the genus level. NucleoSpin^®^Tissue kit (Macherey-Nagel, Düren, Germany), a commercial kit for DNA extraction, was used. DNA quality and quantity were checked in a spectrophotometer Nanodrop^®^ ND-1000 (NanoDrop Technologies/Thermo Scientific, Wilmington, DE, USA). The entire Internal Transcribed Spacer (ITS) (ITS1, 5.8S rDNA gene and ITS2) was amplified using the forward primer NC5 (5’-GTA GGT GAA CCT GCG GAA GGA TCA TT-3’) and the reverse primer NC2 (5’-TTA GTT TCT TTT CCT CCG CT-3’) [32]. PCR assays were carried out in a total volume of 25 µL containing 100 ng of genomic DNA, 0.3 µM of each primer, 2.5 µL of 10× buffer, 1.5 mM of MgCl_2_, 0.2 mM of Deoxynucleotide Triphosphates (dNTPs) and 0.625 U of Taq DNA polymerase (Roche Diagnostics GmbH, Mannheim, Germany). PCR cycling parameters included denaturation at 94 °C for 2 min, followed by 35 cycles of 94 °C for 30 s, annealing at 55 °C for 30 s, extension at 72 °C for 75 s, and a final extension at 72 °C for 7 min. PCR products were purified for sequencing using ExoSAP-IT ^©^ (US Biochemical, Cleveland, OH, USA) following the manufacturer’s protocols. Sequencing was performed by Secugen (Madrid, Spain) and the chromatograms were analyzed using the program ChromasPro version 1.41 Technelysium Pty Ltd. (Unit 406, 8 Cordelia St, South Brisbane QLD 4101, Australia). All sequences were searched for similarity using Basic Local Alignment Search Tool (BLAST) through web servers of the National Center for Biotechnology Information (USA).

### 2.4. Statistical Analysis

The total number of larvae (N), mean number of larvae per product (MN ± SD), frequency of products containing at least 1 larva (F), and percentage of parasite species per product (P) were evaluated. Statistical significance between the MN of different fishing areas and different fish species was performed using the Mann–Whitney test. The differences in F were assessed using the two-sided chi-square test. Concerning N, the differences between AN and SA, between ANM and ANA, between SAM and SAA, and between ANA and SAA were analyzed by the Mann–Whitney test. Statistical differences between parasite species found in AN and SA were evaluated by the Mann–Whitney test (MedCalc for Windows, version 18.11.3-MedCalc Software, Ostend, Belgium). For all tests, *p* < 0.05 was considered significant.

## 3. Results

A total of 30 larvae were collected. In particular, 17 and 13 larvae in AN and SA were found, respectively. Regarding viability, all larvae were considered nonviable or dead because of lack of movement after stimulation. All larvae were morphologically identified as *Anisakis* spp. and then molecularly identified as *A. pegreffii* (30%) and *A. simplex* (70%) (Table 2) [33].

In AN, 70.58% and 29.41% of larvae were identified as *A. simplex* and *A. pegreffii*, respectively. In SA, 69.23% and 30.76% of larvae were classified as *A. simplex* and *A. pegreffii*, respectively. No significant differences (χ^2^ = 0.0065; *p* > 0.05) in the percentages of parasite species between samples were found. Concerning AN, no significant statistical differences were found between *A. simplex* and *A. pegreffii* (*p* > 0.05; U = 30). Concerning SA, no significant statistical differences were found between *A. simplex* and *A. pegreffii* (*p* > 0.05; U = 18). Table 3 shows the parasite infection indexes.

AN recorded higher F, with MN (± SD) values of 0.37 (± 0.64) and 0.28 (± 0.62) larvae in products made of AN and SA found, respectively. Regarding the fish sample species, no significant statistical differences in N (*p* = 0.509; U = 930.5) between the AN and SA samples were found. ANM and SAM were shown to be parasite free. Regarding the Atlantic samples (ANA and SAA), no significant differences in N (*p* = 0.67; U = 192.5) were found. 

## 4. Discussion

The results of the present work showed that deboned, in-oil fillets of anchovy and sardine infected by nonviable *Anisakis* spp. larvae can be found on the market. 

Our results (N and MN) agreed with Guardone et al. [6] who, in a similar trial, found a mean number (MA) of 0.709 larvae per sample of anchovy fillets in oil. The same author reported a significant difference between all categories of product analyzed (salted, in-oil, and marinated anchovies) in regard to number of larvae per product, frequency of contaminated products, and density of larvae per gram. In our study, lower N and MN percentages were probably due to processing techniques; even if fish are previously treated as whole salted anchovies and/or sardines for the maturation process, they are subsequently degutted, deboned, filleted, and put in oil. Beheading/partial gutting carried out by properly trained people during desalting and filleting ensures a lower presence of parasites in these kinds of products. Moreover, the low N, F, and MN values of parasites in AN and SA showed that postmortem larval migration can be prevented by the correct choice of raw materials, application of good manufacturing practices (GMP) such as maintaining cold chains, and by rapid postharvest processing [34,35]. Food business operators (FBO) must ensure that no obviously infected fish or cephalopods join the market. As stated by “Guidance document on the implementation of certain provisions of Regulation (EC) No 853/2004 on the hygiene of food of animal origin” [36], a fishery product is considered to be obviously contaminated if visible parasites are detected in edible portions with no indication regarding the maximum number of parasites allowed in order to discriminate between fit and unfit products [21]. In such a case, FBO must apply their own critical limit to define marketability. In [37], indices such as mean abundance (MA) were used to assess the possibility of selling fish lots and set the MA at 0.3 larvae per sample of fresh product as a threshold limit. The complete elimination of parasites from fishery products is not possible [38]; a Hazard Analysis and Critical Control Point (HACCP)-based program can only minimize the probability or risk of food hazards from occurring. During visual inspection and processing, if edible parts are obviously infected with visible parasites, FBO must apply adequate corrective measures, such as trimming, to ensure that products are “no longer obviously contaminated” [36] with parasites when inspected with the naked eye, and therefore fit for human consumption. If corrective measures turn out to be inadequate, the FBO should not be able to use the product for further processing. For this reason, the application of an efficient sampling method using different parameters could be applied. Concerning this, the use of prediction schemes, as suggested by the European Food Safety Authority [39], like the SADE method [40] for the evaluation of parasite larvae in the edible portion of fish lots, could be useful in terms of food safety and help to reduce economic losses due to consumer rejection. In [41], the SADE scheme was applied to 33 different frozen fish batches, finding that the method, despite the medium acceptable sensitivity, had very high specificity and accuracy, which could possibly allow FBO to assign different purposes to obviously contaminated fish lots. Concerning the potentially hazardous effects related to *Anisakidae* larvae in semipreserved anchovies, the results of this study confirm that the salting process is an effective treatment for the devitalization of *Anisakis* larvae, in agreement with the findings of [2]. In this way, dead larvae should not be considered a risk but rather a defect [19], since they may lead to consumers rejecting the product. Moreover, the presence of larvae, in addition to consumer rejection, may also cause damage to the commercial brand. Even if the potential of nonviable larvae to induce allergies is still under discussion [17,18], taking into account different population sensitization rates, the presence of parasites must be considered a risk only for allergic consumers. Also, according to EFSA, *A. simplex* is, so far, the only fishery product-associated parasite which can cause a clinical allergic response [11]. As far as the provenance of the fish is concerned, in our findings, the prevalence of infection was higher in the Atlantic than in the Mediterranean samples, supporting the findings of [6], who found a higher MN (3.33) in Atlantic samples than in Mediterranean samples (0.42). Rello [42] suggested that a lower frequency of intermediate hosts in the Mediterranean Sea and the higher presence of definitive hosts in the Atlantic areas allows for the maintenance and continuation of the *Anisakis* lifecycle in the Atlantic area [43].

As stated by [11], “for wild-caught fishery products eaten raw or almost raw, information on the prevalence, abundance, as well as species and geographical distributions of the parasites and their hosts together with monitoring systems and trends in parasite presence and abundance are important.” All larvae found in this study were deeply embedded in the flesh, and 70% were *A. simplex* larvae. These results agreed with [44], who noted that, in *M. merluccius* fished in the Atlantic area, *A. simplex* larvae exceeded *A. pegreffii* larvae in the flesh of the same fish host due to the higher flesh penetration rate of *A. simplex* [33]. Finally, the results of our study support the assumptions of other authors [27,45] that geographical area is an important factor to consider in a risk analysis.

## 5. Conclusions

Based on our results, FBO processing fish at industrial or artisanal levels should include risk management measures in their self-checking programs, taking into account the origins of the raw materials in supplier evaluation procedures; it was further stated in [46] that “Visual inspection shall be performed on a representative number of samples. The persons in charge of establishments shall determine the scale and frequency of the inspections by reference to the type of fishery products, their geographical origin and their use.” Moreover, due to the recognized fact that processing technologies can influence the presence and viability of parasites in final products, FBO should implement a system during production whereby visual inspection of eviscerated fish must be carried out by trained personnel [47], particularly in the case of mechanical evisceration, by sampling a representative number of samples not less than 10 fish per batch, as stated by [45]. This step is of strategic significance, since the results of visual inspection and the assessment of the prevalence of larvae would indicate the most appropriate kind of processing (salting, preparation in oil, or marinating), thereby helping to avoid economic losses, withdrawal and recall, RASFF notification, and brand damage. Development of common modus operandi in sampling procedures at the EU level and homogeneous corrective measures are advisable.

## Figures and Tables

**Table 1 animals-10-01807-t001:** Summary of anchovy (AN) and sardine (SA) products, brands, numbers of samples, Food and Agriculture Organization geographical (FAO) origins, gross weights, and sites of collection.

Brand	Anchovy Products (Code)	Geographical Origin (label)	Gross Weight (g)	Sardine Products (Code)	Geographical Origin (Label)	Gross Weight (g)	EU Retailers
1	1ANa, 1ANb	Mar Ionio: FAO 37.2.2	80	1SAa, 1SAb, 1SAc	Mediterranean Sea	100	Supermarket in Torino
2	2ANa, 2ANb	Mediterranean Sea	80	2SAa, 2SAb, 2SAc	Mediterranean Sea	100	Supermarket in Torino
3	3ANa, 3ANb	Tunisia	80	3SAa, 3SAb	Mediterranean Sea	100	Supermarket in Battipaglia
4	4ANa, 4ANb	Mediterranean Sea	80	4SAa, 4SAb	Albania	100	Supermarket in Battipaglia
5	5ANa, 5ANb	Tunisia	80	5SAa, 5SAb	Marocco: FAO 34	100	Supermarket in Battipaglia
6	6ANa, 6ANb	Campania Region	78	6SAa, 6SAb	Marocco: FAO 34	120	Supermarket in Battipaglia
7	7ANa, 7ANb	Tunisia	80	7SAa, 7SAb	Tunisia	100	Supermarket in Torino
8	8ANa, 8ANb	Albania	80	8SAa, 8SAb	Tunisia	100	Supermarket in Torino
9	9ANa, 9ANb	Marocco: FAO 34	80	9SAa, 9SAb	Mediterranean Sea	100	Supermarket in Torino
10	10ANa, 10ANb	Marocco: FAO 34	80	10SAa, 10SAb	Mediterranean Sea	100	Supermarket in Battipaglia
11	11ANa, 11ANb	Albania	58	11SAa, 11SAb	Mediterranean Sea	100	Supermarket in Battipaglia
12	12ANa, 12ANb	Atlantic NE: FAO 27.8 C	70	12SAa, 12SAb	Atlantic NE	100	Supermarket in Battipaglia
13	13ANa, 13ANb	Cantabria: Atlantic NE	45	13SAa, 13SAb, 13SAc	Atlantic NE	120	Local market in Barcelona
14	14ANa, 14ANb	Cantabria: Atlantic NE	45	14SAa, 14SAb, 14SAc	Atlantic NE	100	Local market in Barcelona
15	15ANa, 15ANb	Cantabria: Atlantic NE	58	15SAa, 15SAb, 15SAc	Mediterranean Sea	125	Local market in Barcelona
16	16ANa, 16ANb	Cantabria: Atlantic NE	58	166SAa, 16SAb, 16SAc	Mediterranean Sea	125	Supermarket in Barcelona
17	17ANa, 17ANb	Atlantic NE	80	17SAa, 17SAb, 17SAc	Atlantic NE	125	Supermarket in Barcelona
18	18ANa, 18ANb	Atlantic NE	80	18SAa, 18SAb	Atlantic NE	120	Supermarket in Ibiza
19	19ANa, 19ANb	Atlantic NE	80	19SAa, 19SAb	Atlantic NE	100	Supermarket in Ibiza
20	20ANa, 20ANb	Atlantic NE	90				Supermarket in Ibiza
21	21ANa	Tunisia	80				Supermarket in Barcelona
22	22ANa	Tunisia	80				Supermarket in Torino
23	23ANa	Mediterranean Sea	80				Supermarket in Barcelona
24	24ANa	Mediterranean Sea	90				Supermarket in Torino
25	25ANa	Mediterranean Sea	80				Supermarket in Ibiza

**Table 2 animals-10-01807-t002:** Identified parasites in fish products. Accession ID related to the aligned sequences and web links (https://www.ncbi.nlm.nih.gov/pubmed/).

Identified Parasites	Accession ID	Web Link	Products Code
*A. simplex*	EU624342.1	https://www.ncbi.nlm.nih.gov/nuccore/EU624342.1	12ANa
*A. simplex*	JX237370.1	https://www.ncbi.nlm.nih.gov/nuccore/403492157/	13ANa
*A. simplex*	JX237370.1	https://www.ncbi.nlm.nih.gov/nuccore/403492157/	13ANa
*A. pegreffii*	KF032066.1	https://www.ncbi.nlm.nih.gov/nuccore/KF032066.1	14ANb
*A. simplex*	JX237370.1	https://www.ncbi.nlm.nih.gov/nuccore/403492157/	14ANb
*A. simplex*	JX237370.1	https://www.ncbi.nlm.nih.gov/nuccore/403492157/	15ANa
*A. simplex*	JX237370.1	https://www.ncbi.nlm.nih.gov/nuccore/403492157/	15ANa
*A. simplex*	JX237370.1	https://www.ncbi.nlm.nih.gov/nuccore/403492157/	16ANa
*A. pegreffii*	KF032066.1	https://www.ncbi.nlm.nih.gov/nuccore/KF032066.1	16ANb
*A. simplex*	JX237370.1	https://www.ncbi.nlm.nih.gov/nuccore/403492157/	17ANa
*A. simplex*	JN968834.1	https://www.ncbi.nlm.nih.gov/nuccore/JN968834.1	17ANb
*A. simplex*	JX237370.1	https://www.ncbi.nlm.nih.gov/nuccore/403492157/	18ANa
*A. pegreffii*	KF032066.1	https://www.ncbi.nlm.nih.gov/nuccore/KF032066.1	18ANa
*A. pegreffii*	KF032066.1	https://www.ncbi.nlm.nih.gov/nuccore/KF032066.1	18ANb
*A. simplex*	GQ169362.1	https://www.ncbi.nlm.nih.gov/nuccore/GQ169362.1	19ANa
*A. simplex*	JX237370.1	https://www.ncbi.nlm.nih.gov/nuccore/403492157/	19ANb
*A. pegreffii*	KF032066.1	https://www.ncbi.nlm.nih.gov/nuccore/KF032066.1	20ANa
*A. simplex*	GQ169362.1	https://www.ncbi.nlm.nih.gov/nuccore/GQ169362.1	13SAa
*A. simplex*	GQ169362.1	https://www.ncbi.nlm.nih.gov/nuccore/GQ169362.1	13SAa
*A. pegreffii*	KF032066.1	https://www.ncbi.nlm.nih.gov/nuccore/KF032066.1	13SAb
*A. pegreffii*	KF032066.1	https://www.ncbi.nlm.nih.gov/nuccore/KF032066.1	13SAc
*A. simplex*	JX237370.1	https://www.ncbi.nlm.nih.gov/nuccore/403492157/	14SAa
*A. simplex*	JN968834.1	https://www.ncbi.nlm.nih.gov/nuccore/JN968834.1	14SAa
*A. simplex*	GQ169362.1	https://www.ncbi.nlm.nih.gov/nuccore/GQ169362.1	17SAb
*A. pegreffii*	KF032066.1	https://www.ncbi.nlm.nih.gov/nuccore/KF032066.1	17SAb
*A. pegreffii*	KF032066.1	https://www.ncbi.nlm.nih.gov/nuccore/KF032066.1	17SAc
*A. simplex*	JX237370.1	https://www.ncbi.nlm.nih.gov/nuccore/403492157/	18SAa
*A. simplex*	JN968834.1	https://www.ncbi.nlm.nih.gov/nuccore/JN968834.1	18SAb
*A. simplex*	EU624342.1	https://www.ncbi.nlm.nih.gov/nuccore/EU624342.1	18SAb
*A. simplex*	GQ169362.1	https://www.ncbi.nlm.nih.gov/nuccore/GQ169362.1	19SAb

**Table 3 animals-10-01807-t003:** Results divided per product type: Total anchovy samples (AN), total sardine samples (SA), Mediterranean anchovies (ANM), Mediterranean sardines (SAM), Atlantic anchovies (ANA), and Atlantic sardines (SAA). Number of analyzed products, number of products with at least 1 larva, number of collected larvae, and frequency of products with at least 1 larva (%). MN: mean number of larvae per product type; SD: standard deviation.

Type	Analyzed Products (n)	Products Containing 1 Larva at Least (n)	Parasites Found (n)	Frequency of Products with 1 Larva at Least (%)	MN ± SD
AN	45	13	17	28.88	0.37 ± 0.64
SA	45	9	13	20	0.28 ± 0.62
ANM	23	0	0	0	0
SAM	26	0	0	0	0
ANA	22	13	17	59.09	0.75 ± 0.75
SAA	19	9	13	47.36	0.66 ± 0.82

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
