# Peer review of "Anisakis spp. Larvae in Deboned, in-Oil Fillets Made of Anchovies (Engraulis encrasicolus) and Sardines (Sardina pilchardus) Sold in EU Retailers"

_animals, 2020, doi:10.3390/ani10101807_

Round 1

Reviewer 1 Report

The authors produced an interesting and useful paper. In my opinion it well written and informative, and may be published as it is after correction 3 misspellings ( lines 38 and 39 - italicize Anisakis; line 76 delete (?) n. before 39.

Author Response

REBUTTAL LETTER ANIMALS - 953877

REVISOR 1

The authors produced an interesting and useful paper.

In my opinion it well written and informative, and may be published as it is after correction 3 misspellings (lines 38 and 39 - italicize Anisakis; line 76 delete (?) n. before 39.

As suggested by revisor, Anisakis was corrected in italics (lines 38 and 39, page 2) and “n.” before 39 was deleted (line 76, page 2)

Reviewer 2 Report

The manuscript describes the presence of Anisakis spp. larvae in deboned anchovies and sardine fillets in oil marketed in the EU in order to assess the influence of processing techniques on the prevalence of larvae. The paper is very interesting and can be accepted with minor spelling revision.

Lines 1: modify “Larvae” in “larvae”

Line 25: “90 semi-preserved” modify in “Ninety semi-preserved”

Line 38 and 39: “Anisakis spp.” modify in “Anisakis spp.”

Line 40: “90 semi-preserved” modify in “Ninety semi-preserved”

Line 41: “The Mediterranean or The Atlantic Ocean” modify in “the Mediterranean Sea or the Atlantic Ocean”

Line 43: “30…” modify in “Thirty”

Line 48: “Food safety” modify in “food safety”, or uniform also all the others keywords

Line 61: “kg. [1].” remove the point between kg and the [.

Line 68: “stable because of the low activity water low moisture and” modify in “stable because of the low activity water, low moisture and”

Line 67-70: please rewrite this sentence, is very long and very complicated to understand.

Line 81-86: regarding the reported effects of Anisakis, the following recent article could be discussed by the authors. “Speciale, A., Trombetta, D., Saija, A., ...Cimino, F., Gangemi, S. (2017). Exposure to Anisakis extracts can induce inflammation on in vitro cultured human colonic cells. Parasitology Research, 116(9), pp.2471-2477”

Line 90: Table 1: modify all the “Mediterranean sea” in “Mediterranean Sea”

Lines 102-103: instead of “Mediterranean area” is better “Mediterranean Sea” and “The Atlantic Ocean” modify in” the Atlantic Ocean”.

Line 148: “All larvae were morphologically identified” insert a reference for the methods used such as “Rie Murata, Jun Suzuki, Kenji Sadamasu, Akemi Kai (2011). Morphological and molecular characterization of Anisakis larvae (Nematoda: Anisakidae) in Beryx splendens from Japanese waters. Parasitol Int . 2011 Jun;60(2):193-8. doi: 10.1016/j.parint.2011.02.008. Epub 2011 Mar 3.” or other methods; and eventually insert the type I or II if done.

AFTER LINE 152 no more number line are present and the page number are not correct.

?? Line 157: “30); Concerning SA” please verify or put a “.” or use not the capital letter

?? Line 158: “18); Table” same problem

Line ???: “Regarding the Atlantic sampleS” please verify

Line ???: “Our results (N and MA) were in agreement with [6]” insert at least the name of the author…

Line ???: “Mediterranean ones (0.42). [40] suggest” please verify

Line ???+1: “a lower frequency of intermediate hosts in The Mediterranean Sea” please verify “…the Mediterranean Sea….

Conclusion

Line xxx: “evaluation procedure the origin of the raw material. [44] also stated that” please verify there is a point between material and [

Author Response

REBUTTAL LETTER ANIMALS - 953877

REVISOR 2

The manuscript describes the presence of Anisakis spp. larvae in deboned anchovies and sardine fillets in oil marketed in the EU in order to assess the influence of processing techniques on the prevalence of larvae. The paper is very interesting and can be accepted with minor spelling revision.

Lines 1: modify “Larvae” in “larvae”

Line 2, page 1: as suggested by revisor, “Larvae” was modified in “larvae”

Line 25: “90 semi-preserved” modify in “Ninety semi-preserved”

Line 25, page 2: as suggested by revisor, “90” was replaced with “ninety”

Line 38 and 39: “Anisakis spp.” modify in “Anisakis spp.”

Line 38 and 39, page 2: As suggested by revisor, Anisakis was corrected in italics

Line 40: “90 semi-preserved” modify in “Ninety semi-preserved”

Line 40, page 2: as suggested by revisor, “90” was replaced with “ninety”

Line 41: “The Mediterranean or The Atlantic Ocean” modify in “the Mediterranean Sea or the Atlantic Ocean”

Line 41, page 2: as suggested by revisor, “The” was replaced with “the” deleting capital letters

Line 43: “30…” modify in “Thirty”

Line 43, page 2: as suggested by revisor, “30” was replaced with “thirty”

Line 48: “Food safety” modify in “food safety”, or uniform also all the others keywords

Line 48, page 2: as suggested by revisor, “Food safety” was modified in “food safety”

Line 61: “kg. [1].” remove the point between kg and the [.

Line 61, page: as suggested by revisor, the point between kg and the [. was removed

Line 68: “stable because of the low activity water low moisture and” modify in “stable because of the low activity water, low moisture and”

Line 68, page 2: as suggested by revisor, the sentence has been changed

Line 67-70: please rewrite this sentence, is very long and very complicated to understand.

Line 67-70, page 2: as suggested by revisor, the sentence has been changed

Line 81-86: regarding the reported effects of Anisakis, the following recent article could be discussed by the authors. “Speciale, A., Trombetta, D., Saija, A., ...Cimino, F., Gangemi, S. (2017). Exposure to Anisakis extracts can induce inflammation on in vitro cultured human colonic cells. Parasitology Research, 116(9), pp.2471-2477”

Lines 81-86, page 3: as suggested by revisor, reference was added

Line 90: Table 1: modify all the “Mediterranean sea” in “Mediterranean Sea”

Line 99, page 4: as suggested by revisor, all “Mediterranean sea” were changed in “Mediterranean Sea” within the table n. 1

Lines 102-103: instead of “Mediterranean area” is better “Mediterranean Sea” and “The Atlantic Ocean” modify in” the Atlantic Ocean”.

Lines 102-103, page 7: as suggested by revisor, “Mediterranean area” was changed in “Mediterranean Sea”  and “The Atlantic Ocean” was changed in “the Atlantic Ocean”

Line 148: “All larvae were morphologically identified” insert a reference for the methods used such as “Rie Murata, Jun Suzuki, Kenji Sadamasu, Akemi Kai (2011). Morphological and molecular characterization of Anisakis larvae (Nematoda: Anisakidae) in Beryx splendens from Japanese waters. Parasitol Int . 2011 Jun;60(2):193-8. doi: 10.1016/j.parint.2011.02.008. Epub 2011 Mar 3.” or other methods; and eventually insert the type I or II if done.

 Line 148, page 8: as suggested by revisor, reference was added

AFTER LINE 152 no more number line are present and the page number are not correct.

We apoligize for the inconvenience: lines number and pages lines were added

?? Line 157: “30); Concerning SA” please verify or put a “.” or use not the capital letter

Line 155, page 11: as suggested by revisor, a ”.” was added

?? Line 158: “18); Table” same problem

Line 156, page 11: as suggested by revisor, a ”.” was added

Line ???: “Regarding the Atlantic sampleS” please verify

Line 167, page 11: as suggested by revisor, “sampleS” was chenged in “samples”

Line ???: “Our results (N and MA) were in agreement with [6]” insert at least the name of the author…

Line 172, page 11: as suggested by revisor, name of the author was added (see also line 87, page 3)

Line ???: “Mediterranean ones (0.42). [40] suggest” please verify

Line 217, page 12: as suggested by revisor, name of the author was added

Line ???+1: “a lower frequency of intermediate hosts in The Mediterranean Sea” please verify “…the Mediterranean Sea….

 Line 218, page 12: as suggested by revisor, “The” was chenged in “the”

Conclusion

Line xxx: “evaluation procedure the origin of the raw material. [44] also stated that” please verify there is a point between material and [

Line 233, page 13: as suggested by revisor, sentences was checked and modified